# Oligoclonal IgG antibodies in multiple sclerosis target patient-specific peptides

**Michael Graner[1], Tiffany Pointon[2], Sean Manton[2], Miyoko Green[2], Kathryn Dennison[2], Mollie Davis[2], Gino Braiotta[2], Julia Craft[1], Taylor Edwards[2], Bailey Polonsky[2], Anthony Fringuello[1], Timothy Vollmer[2], Xiaoli Yu [1]***

**1** Department of Neurosurgery, University of Colorado, Anschutz Medical Campus, Aurora, Colorado, United States of America, **2** Department of Neurology, University of Colorado, Anschutz Medical Campus, Aurora, Colorado, United States of America

* Xiaoli.Yu@CUanschutz.edu

**Data Availability Statement:** All relevant data are within the paper and its Supporting Information files.

**Funding:** XY received Research Award (RG 3934A1/1) from National Multiple Sclerosis Society, https://www.nationalmssociety.org/. The

## Abstract

IgG oligoclonal bands (OCBs) are present in the cerebrospinal fluid (CSF) of more than 95% of patients with multiple sclerosis (MS), and are considered to be the immunological hallmark of disease. However, the target specificities of the IgG in MS OCBs have remained undiscovered. Nevertheless, evidence that OCBs are associated with increased levels of disease activity and disability support their probable pathological role in MS. We investigated the antigen specificity of individual MS CSF IgG from 20 OCB-positive patients and identified 40 unique peptides by panning phage-displayed random peptide libraries. Utilizing our unique techniques of phage-mediated real-time Immuno-PCR and phage-probed isoelectric focusing immunoblots, we demonstrated that these peptides were targeted by intrathecal oligoclonal IgG antibodies of IgG1 and IgG3 subclasses. In addition, we showed that these peptides represent epitopes sharing sequence homologies with proteins of viral origin, and proteins involved in cell stress, apoptosis, and inflammatory processes. Although homologous peptides were found within individual patients, no shared peptide sequences were found among any of the 42 MS and 13 inflammatory CSF control specimens. The distinct sets of oligoclonal IgG-reactive peptides identified by individual MS CSF suggest that the elevated intrathecal antibodies may target patient-specific antigens.

## Introduction

Multiple sclerosis (MS) is the most common autoimmune disease of the central nervous system (CNS) and is characterized by inflammatory demyelination and neuronal damage. Cerebrospinal fluid (CSF) oligoclonal bands (OCBs), a characteristic feature of MS, are associated with increased levels of disease activity and disability [1–3], the conversion from a clinically isolated syndrome to early relapsing-remitting MS [4], greater brain atrophy [5], and increased cortical lesion load/intrathecal inflammation [6]. Furthermore, recent studies have shown that antibodies produced by clonally expanded plasma cells in MS CSF cause demyelination [7], and myelin-specific MS antibodies cause complement-dependent oligodendrocyte loss and demyelination [8]. This evidence supports the notion that intrathecal IgG in MS plays a critical

funder had no role in study design, data collection and analysis, decision to publish, or preparation of the manuscript.

**Competing interests:** The authors have declared that no competing interests exist.

role in disease pathogenesis, consistent with the view that CSF IgG alone remains the best marker of disease activity in individual MS patients [1].

OCBs have been assumed to target antigens relevant to MS pathogenesis, with leading antigen candidates being myelin proteins and/or viruses. However, despite intensive research over the last several decades, the target specificities of the IgG within OCB in MS have remained a mystery. A recent report by Brändle et al. [9] showed that OCBs in MS target ubiquitous intracellular antigens released in cellular debris. We hypothesized that phage-displayed random peptide libraries can be used to identify antigenic peptides specific to intrathecal IgG of MS.

We have previously shown that phage peptides reactive to OCBs are persistent in MS patients [10], suggesting that these peptides can be used as unique tools for investigating the specificity of OCBs and to investigate disease pathogenesis. To further elucidate OCBs specificity in MS, we used 20 OCB-positive MS CSF IgG to screen phage-displayed random peptide libraries and identified 40 high-affinity peptides which were reactive to intrathecal oligoclonal IgG in most MS patients. We also show that these peptide antigens are unique in each patient. Our data suggest that the oligoclonal bands in MS may target patient-specific antigens.

## Materials and methods

### Patients

With approval of the University of Colorado Institutional Review Board (COMIRB # 00–688), CSF and sera from MS patients and controls were collected at University of Colorado Hospital after obtaining written consent CSFs were immediately centrifuged at 500 x g for 10 minutes, and the supernatant was collected. Both CSF and sera were stored at −80˚C until use. The CSF of all MS patients contained oligoclonal bands (OCBs, determined by ARUP Laboratories, SLC, UT). CSF IgG concentration, percent of IgG in CSF, and number of OCBs from each patient are listed in Table 1.

### Identification of high affinity phage peptides with MS CSF IgG

Ph.D.-7™ and Ph.D.-12™ Phage Display Peptide Library (New England BioLabs, Beverly, MA) kits were used for affinity selection of specific peptides by all MS CSF. The Ph.D.-12 library is a combinatorial library of random 12-mer peptides fused to a minor coat protein (pIII) of M13 phage. The displayed peptide is expressed at the N-terminus of pIII. The library consists of approximately $10^9$ electroporated (i.e., unique) sequences. Similarly, the Ph.D.-7 library is a combinatorial library consisted of $10^9$ unique random heptapeptides. The panning procedure as well as characterization of positive phage peptides were as described [10]. A streamlined protocol was used to determine phage peptide specificity after affinity selection [10]. Briefly, individual phage plaques were amplified in U96 DeepWell plates and used to determine reactivity to panning MS CSF IgG by 96-well ELISA [11]. Positive clones were confirmed by duplicate phage ELISA with a pre-immune human IgG control. DNA from positive phage clones were purified and sequenced.

### Dose–response phage-mediated real-time Immuno-PCR

Phage-mediated real-time Immuno-PCR (IPCR) was performed as described [12]. Reacti-Bind™ wells of protein A-coated clear strip plates (Thermo Scientific, Rockford, IL) were coated with 50 µl of CSF or serum (1 µg/ml IgG) and with pre-immune human IgG (Alpha Diagnostic) in TBS (50 mM Tris–HCl, 150 mM NaCl) at room temperature for two hours, washed with TBS containing 0.05% Tween 20 (TBST), and blocked with 3% nonfat dry milk/ 0.05% TBST at room temperature for one hour. Serial 10-fold phage dilutions in duplicate

**Table 1. Clinical characteristics of MS patients.**

| Patient # | Patient ID | Sex | CSF IgG (µg/mL) | % IgG | OCBs | Diagnosis |
|---|---|---|---|---|---|---|
| MS #1 | MS 02–19 | F | 68 | 15.00 | 6 | PPMS |
| MS #2 | MS 02–21 | F | 65 | 16.00 | 3 | RRMS |
| MS #3 | MS 02–24 | F | 161 | 16.90 | 1 | SPMS |
| MS #4 | MS 03–01 | F | 70 | 23.00 | 6 | RRMS |
| MS #5 | MS 03–07 | F | 86 | 32.00 | 3 | RRMS |
| MS #6 | MS 04–02 | F | 219 | 21.00 | 5 | PPMS |
| MS #7 | MS 04–03 | M | 92 | 17.60 | 2 | RRMS |
| MS #8 | MS 04–05 | F | 46 | 16.40 | + | RRMS |
| MS #9 | MS 04–07 | F | 31 | 14.10 | + | PPMS |
| MS #10 | MS 05–01 | M | 92 | 23.00 | + | RRMS |
| MS #11 | MS 05–02 | F | 72 | 24.80 | 19 | RRMS |
| MS #12 | MS 05–03 | F | 57 | 23.70 | 21 | RRMS |
| MS #13 | MS 05–04 | F | 68 | 17.40 | 22 | SPMS |
| MS #14 | MS 05–06 | M | 28 | 8.00 | 14 | PPMS |
| MS #15 | MS 05–07 | M | 98 | 18.40 | 8 | RRMS |
| MS #16 | MS 05–08 | F | 112 | 22.40 | 19 | RRMS |
| MS #17 | MS 05–10 | F | 88 | 20.90 | 28 | RRMS |
| MS #18 | MS 06–02 | F | 12 | 5.50 | 12 | RRMS |
| MS #19 | MS 06–03 | F | 32 | 9.40 | 13 | RRMS |
| MS #20 | MS 06–06 | F | 127 | 33.40 | 19 | RRMS |

All MS patients whose CSF were used for panning phage-displayed random peptide libraries are included. Major immunological features such as CSF IgG concentration, percent of IgG in the CSF, number of oligoclonal bands and diagnosis are listed. The %IgG is the percent of total protein in CSF that corresponds to IgG. RRMS: relapsing remitting multiple sclerosis; SPMS: secondary progressive multiple sclerosis; PPMS: primary progressive multiple sclerosis.

were added to MS CSF/serum IgG-coated wells and incubated at room temperature for two hours. After washing, bound phage were lysed in 50 µl of double-deionized water by heating the plates at 95 ˚C for 15 minutes. Single-stranded phage DNA was released and used as template for real-time PCR in an Applied Biosystems 7500 Fast Real-Time PCR system (Applied Biosystems, Foster City, CA). For standard SYBR® Green PCR, each reaction (20 µl) consisted of 1× power SYBR® Green master mix (Applied Biosystems), 750 nM of each M13 phage primer and 4 µl of phage template. Thermal cycle conditions were 95 ˚C for 10 minutes, followed by 40 cycles at 95 ˚C for 15 seconds and 60 ˚C for 45 seconds. Fast real-time PCR was conducted using 1× Fast SYBR® Green master mix, with thermal cycling at 95 ˚C for 20 seconds, followed by 40 cycles at 95 ˚C for three seconds and 60 ˚C for 30 seconds. A control reaction without template was included in each run.

## Isoelectric focusing (IEF) immunoblotting

CSF (200–500 µl) was concentrated on an Amicon Ultra 0.5-ml 30 K cellulose centrifugal filter unit at 14,000×g for 30 minutes at room temperature before IEF (SPIFE® IgG IEF kit, Helena Laboratories, Beaumont, TX) using SPIFE 3000 electrophoresis analyzer. Wicks were soaked in an anode (0.3 M acetic acid) or cathode (1 M NaOH) solution and applied to the edge of a SPIFE® IgG IEF gel. Five microliters of concentrated MS CSF/sera (3–5 µg IgG for phage probe and 100 ng IgG for alkaline phosphatase-conjugated anti-human IgG probe) were loaded into wells of an SPIFE IEF gel. After electrophoresis at 700 V for one hour at 15 ˚C, samples were transferred to PVDF membranes for 45 minutes, followed by blocking in Helena

blocking agent (1 g bovine milk protein/50 ml 1× TBS) for one hour at room temperature. Membranes were incubated with the respective phage peptide at concentrations ranging from $5.0×10^{10}$ to $1.5×10^{11}$ pfu/ml in 1:10 Helena blocking agent/TBST (blocking buffer) at room temperature for two hours. After washing with 0.05% Tween-TBS, membranes were incubated with mouse anti-M13 mAb at a 1:500 dilution in blocking buffer, followed by incubation with 1:500 dilution of AP conjugated anti-mouse IgG at room temperature for one hour. Membranes were developed with NBT/BCIP substrate. For control blots, membranes were incubated for one hour with 1:1000 dilutions of AP-anti-human IgG (H+L) in blocking buffer, followed by NBT/BCIP detection.

## Western blots determining phage peptide reactivity to CSF/serum IgG antibodies of IgG1/IgG3 subclasses

Phage peptides ($10^{11}$ pfu/per well) in TBS were denatured and reduced by incubation with 1x protein sample buffer containing β-mercaptoethanol (Pierce Biotechnology, Rockford, IL) at 95 ˚C for 10 minutes, and separated in BioRad 4–15% Tris/Glycine gel for 50 minutes at a constant 200 V. The gels were electro-blotted onto PVDF membranes (Bio-Rad, Hercules, CA) for 60 minutes at a constant 15 V using Trans-Blot® Semi-Dry Cell (Bio-Rad). After blocking for one hour with 1× casein/TBS (Vector Labs, Burlingame, CA) containing 0.1% Tween 20, the blots were incubated with corresponding MS CSF and serum (primary antibodies, at 1 μg/ml) at 4˚C overnight. The bound CSF and serum IgG antibodies were then detected with HRP-mouse anti-human IgG1 and IgG3 antibodies (1:5000) respectively. Isotype-specific mouse monoclonal anti-human IgG1 (I2513, clone 8c/6-39) and anti-human IgG3 (I7260, clone HP-6050) antibodies were used (Sigma). This was followed by secondary antibody anti-mouse IgG (H+L) incubation and detection with SuperSignal® West Femto Maximum Sensitivity chemiluminescent substrate (Pierce Biotechnology).

For detection of phage pIII protein, duplicate membranes were incubated with a 1: 25,000 dilution of mouse anti-M13 pIII mAb (New England BioLabs, Ipswich, MA), followed by a 1: 25,000 dilution of HRP-conjugated goat anti-mouse IgG (Vector Labs) as secondary Ab and with SuperSignal® West Pico substrate for chemiluminescent detection (Pierce Biotechnology).

## Quantifying band intensity of western blots

The FluorChem Q™ system was used to detect the signal produced by addition of chemiluminescent substrate to probed blots. Digital images of the blots were collected by the FluorChem Q at several different lengths of time of exposure to optimize for image clarity and quality. The images were then analyzed quantitatively by AlphaView software for FluorChem™ systems.

## DNA sequencing and database searches

Single-stranded phage DNA was purified and sequenced to deduce the amino acid sequence of the peptides. Consensus peptides were identified by sequence alignment using ClustalW (http://www.ebi.ac.uk/clustalw/). To identify candidate proteins, the most abundant peptides panned by CSF from each patient were searched in BLAST (http://www.ncbi.nlm.nih.gov/) using the Swiss Prot protein sequence database.

## Results

### Specific phage peptides were identified by CSF from 14/20 OCB-positive MS patients, but no common peptide sequences were found

We studied a total of 20 MS patients to investigate peptide antigen specificity of the intrathecal IgG. These patients were all positive for oligoclonal bands. Table 1 lists key clinical immunological features of the patients, including CSF IgG concentration, percent of IgG in the CSF, and the number of oligoclonal bands. We applied Phage-Displayed Random Peptide Libraries technologies (Ph.D.-7™ and Ph.D.-12™, New England Biolab) for a minimum of three rounds of panning with MS CSF for this study. After three to five rounds of affinity selection with each of the 20 MS CSF, phage peptides were analyzed for specificity using our streamlined high throughput protocol as previously described [10]. 14 MS CSF selected positive phage peptides, while the CSF from six MS patients failed to identify any positive phage clones with a repeated panning approach, as well as using additional ultra-fast selection of peptide method [13]. All positive phage clones were amplified, and the phage DNA was purified and sequenced to deduce peptide sequences [11]. A total of 40 unique peptides were identified by each of the 14 MS CSF IgG, ranging from one to six peptides per patient. Although homologous peptides were identified within each MS CSF, no shared peptide sequences were found among MS patients (Table 2).

### Phage peptides target intrathecally synthesized oligoclonal IgG and were recognized by IgG1 and IgG3 subclasses

To determine whether CSF-selected phage peptides were specific for intrathecally synthesized IgG in MS patients, we tested equal amounts of CSF and paired serum IgG (50 ng IgG per well) for peptide binding specificity. We utilized our highly sensitive dose-dependent phage mediated immuno-PCR (phage-IPCR) method [12]. MS serum and CSF were coated onto wells of protein A plates, followed by addition of serial 10-fold dilutions of corresponding phage peptides to each well, and phage binding specificity was assessed by real-time PCR [10]. Phage peptides selected by CSF IgG of all 14 MS patients bound more to CSF IgG than to serum IgG of the same patient in a dose-dependent manner, and there is a significant difference of phage binding between CSF and serum ($p = 0.0002$). Fig 1A and 1B show a representative data of greater binding of phage peptides to CSF IgG than to paired serum IgG in two MS patients (1A: MS 02–19; 1B MS 03–7). Fig 1C is the summary data demonstrating intrathecal IgG binding of the peptides to all 14 MS CSF tested. These results demonstrate that the phage peptides were preferentially targeted by intrathecally-synthesized IgG in MS CSF.

To further demonstrate that the intrathecal IgG-specific phage peptides were recognized by oligoclonal IgG bands, we examined phage binding specificity to MS CSF and paired serum by isoelectric focusing (IEF) immunoblotting using phage peptides as probes. Paired CSF and serum were separated on agarose IEF gels, transferred to PVDF membranes, and probed with corresponding phage peptides [10]. In all seven MS patients, phage peptides were recognized by multiple high-intensity IgG bands in the CSF, while fewer, less intense, or no bands were detected in the paired serum (Fig 2). To confirm that the IgG bands recognized by phage peptides represent oligoclonal bands in CSF, duplicate serum and CSF IEF blots were probed with anti-human IgG and are shown in Fig 2 next to the phage blots for comparison. All peptide-reactive bands in the CSF corresponded to bands of oligoclonal IgG detected by anti-human IgG antibody.

Additionally, we carried out Western blots of purified phage to determine the IgG subclass specificity of these peptides. Phage ($10^{10}$ pfu/well) were separated on a 4–15% SDS-PAGE gel,

**Table 2. Unique peptides were identified from each MS CSF IgG.**

| Patient # | Peptide Sequence | Pt ID | Patient # | Peptide Sequence | Pt ID |
|---|---|---|---|---|---|
| MS #1 | N N L T Q S K F L R L Q | MS02-19* | MS #8 | | MS04-5 |
| | S T L S E S K V N R L L | | MS #9 | K P A N L P P W G G Y S | MS04-7 |
| | N A L T E S K Y V K L L | | MS #10 | S L D P Y Q V R W A R H[1] | MS05-1 |
| | T N T L T P H K L Q M L | | | D N L Y P M H R T G I R | |
| MS #2 | NONE | MS02-21 | | | |
| MS #3 | E F G T F L W[1,2] | MS02-24* | MS #11 | A T L T A A T S G S T V | MS05-2 |
| | K F G T A L W | | MS #12 | I P Y H R F P | MS05-3 |
| | Q F G T F L W | | MS #13 | W G L D N P P | MS05-4* |
| | S F G T A L W | | | A P A H Q I P | |
| MS #4 | NONE | MS03-1 | | A P A H H P P[1,2] | |
| MS #5 | H I D V S R P W R V T G | MS03-7* | | A P P H V M P | |
| | T A Q D I S R P W W F P | | | G P V N M N L | |
| | S L G S K M D I S R P W[1] | | MS #14 | F H L P W M Q | MS05-6* |
| | Q H N V S R P W V L F T | | MS #15 | L I S I S E Q R A A L I | MS05-7 |
| | S V S V G M K P S P R P | | MS #16 | L S P D Y L R W I R L N | MS05-8 |
| | T I M D I S R T W T K V | | | G W T H F D K P I G T L | |
| MS #6 | F S K T E P L S P S W F | MS04-2* | | A R T H F D A P P L W N | |
| | N P V E H W L A V L P T | | MS #17 | NONE | MS05-10 |
| | N N L T Q S K F L R L Q | | MS #18 | NONE | MS06-2 |
| | H W R H W L A D T A F P | | MS #19 | F Y S H S F P P | MS06-3 |
| MS #7 | V L N W H P F[1,2] | MS04-3* | MS #20 | NONE | MS06-6 |
| | M F N W H P F | | | | |

*peptides were published previously [10]. No shared peptide sequences were found between MS patients.

[1]Phage peptides used for western blots in.

[1,2] Phage peptides used for both western blots and for screening MS and IC CSF (Fig A and B in S1 Fig).

blotted, and probed with corresponding CSF and paired serum as primary antibodies, followed by incubation with mouse anti-human IgG1 and IgG3 secondary antibodies. Shown here in Fig 3 are representative blots from two patients MS #7 (MS 04–3), and MS#13 (05–4). Phage

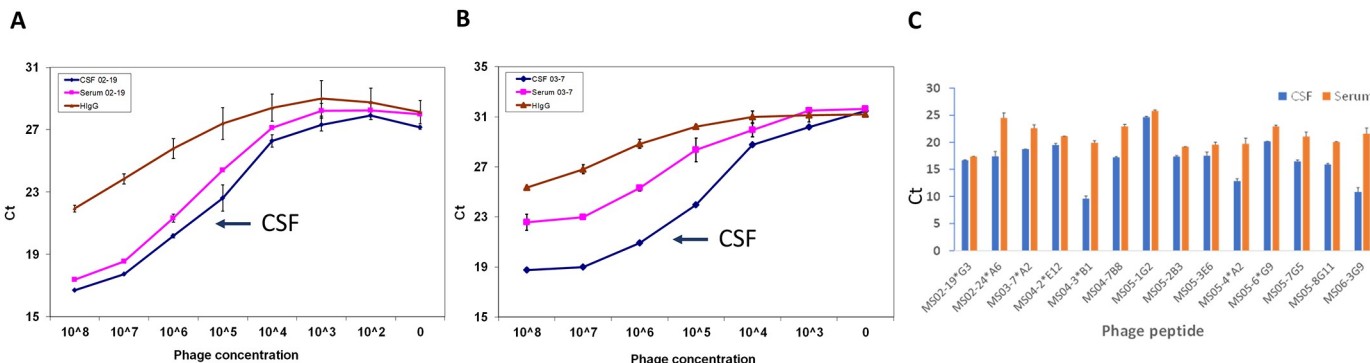

**Fig 1. MS phage peptides target intrathecally synthesized IgG in dose-dependent manner by phage-IPCR.** Representative paired MS serum and CSF, as well as pre-immune human IgG control (50 μl at IgG concentration of 1 μl/ml), were coated in duplicate wells of protein A-plates before addition of the corresponding phage peptides (at serial 10-fold dilutions starting with $10^8$pfu) each well. Bound phage was determined by real-time PCR. Phage peptides bound 5-10-fold higher to MS CSF IgG than to paired serum IgG in a dose-dependent manner. Pre-immune human IgG served as negative control. Error bars represent standard deviation. A, patient 1 (02–19); B, patient 5 (03–7); Data represent at least three independent experiments. C. There is a significant higher binding of phage peptides to CSF than paired serum ($p = 0,0002$, paired Student's T-Test). Phage peptides ($10^5$–$10^8$) were assessed for binding by IPCR as described above.

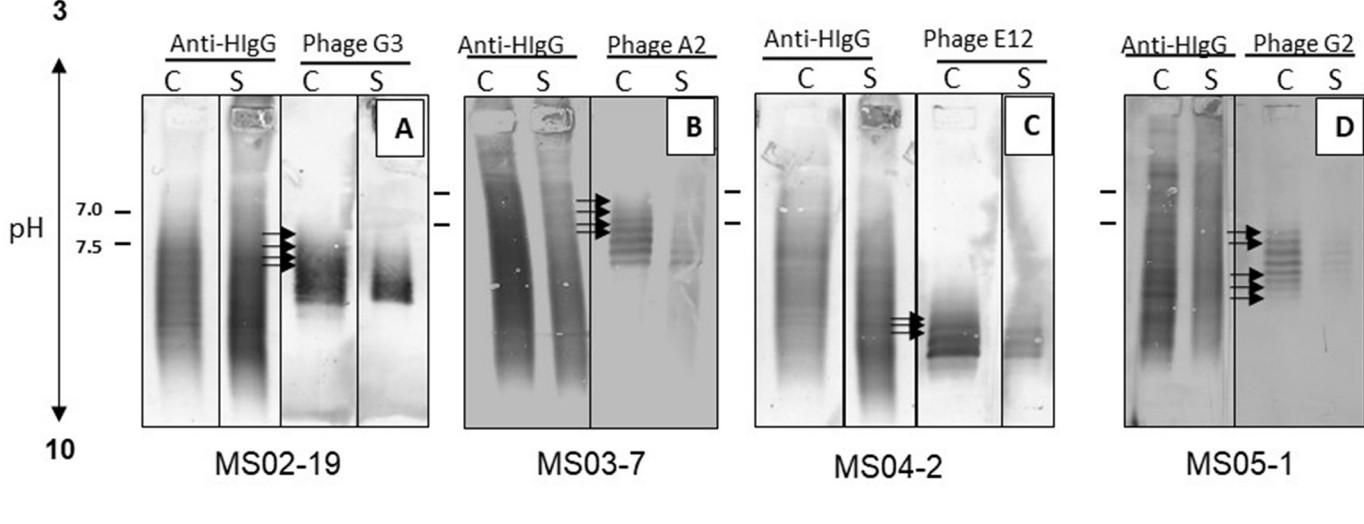

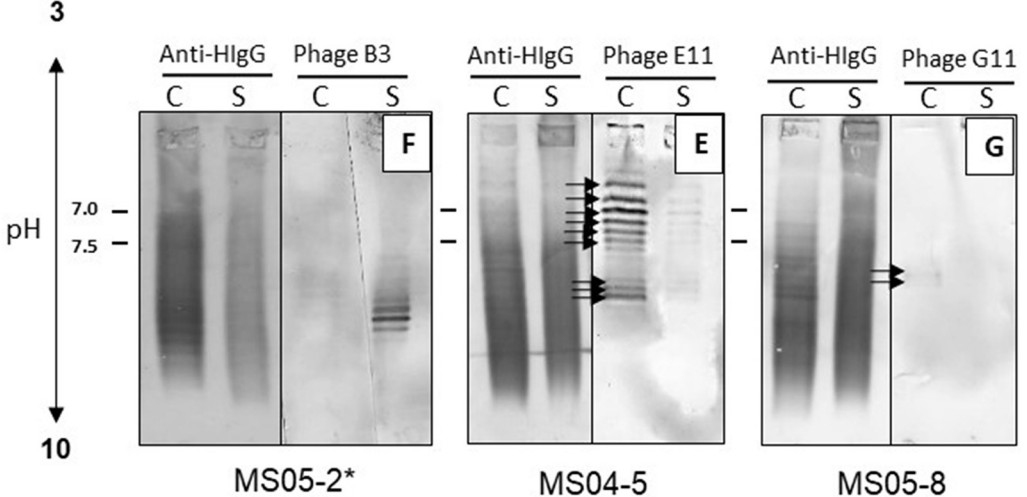

**Fig 2. Phage probed isoelectric focused blots demonstrate that phage peptides were recognized by MS oligoclonal IgG bands.** Paired MS CSF and serum (3–5 µg total IgG) from seven MS patients were resolved on agarose IEF gels and transferred to nitrocellulose membranes. The blots were probed with corresponding phage peptides ($10^{10}$ pfu/ml) and incubated with mouse anti-pIII antibody followed by AP-anti-mouse antibody. Duplicate blots were probed with anti-human IgG as positive controls to reveal total oligoclonal bands. Peptides selected by MS IgG recognized multiple high-density oligoclonal IgG bands in the CSF, but weaker and reduced number of bands in the paired serum. Arrows indicate extra bands detected in the CSF. Patient ID was listed under each blot.

peptides were recognized by both IgG1 and IgG3 subclasses with equal band intensity in both CSF and serum (Fig 3A). Band intensity analysis showed that the ratios of IgG1 band between CSF and serum were comparable as ratios of IgG3 band between CSF and serum (Fig 3B), suggesting that phage peptides are recognized by IgG1 and IgG3 antibodies in both CSF and paired serum of MS patients.

## Screening large number of MS CSF using phage-IPCR did not reveal any shared binding specificity of the intrathecal phage peptides

To determine whether these selected phage peptides shared common antigen bindings to MS intrathecal IgG, we screened 42 MS CSF and 13 inflammatory controls (acute viral meningitis, Behcet's disease, paraneoplastic syndrome, viral meningitis, Cryptococcal meningitis, subacute

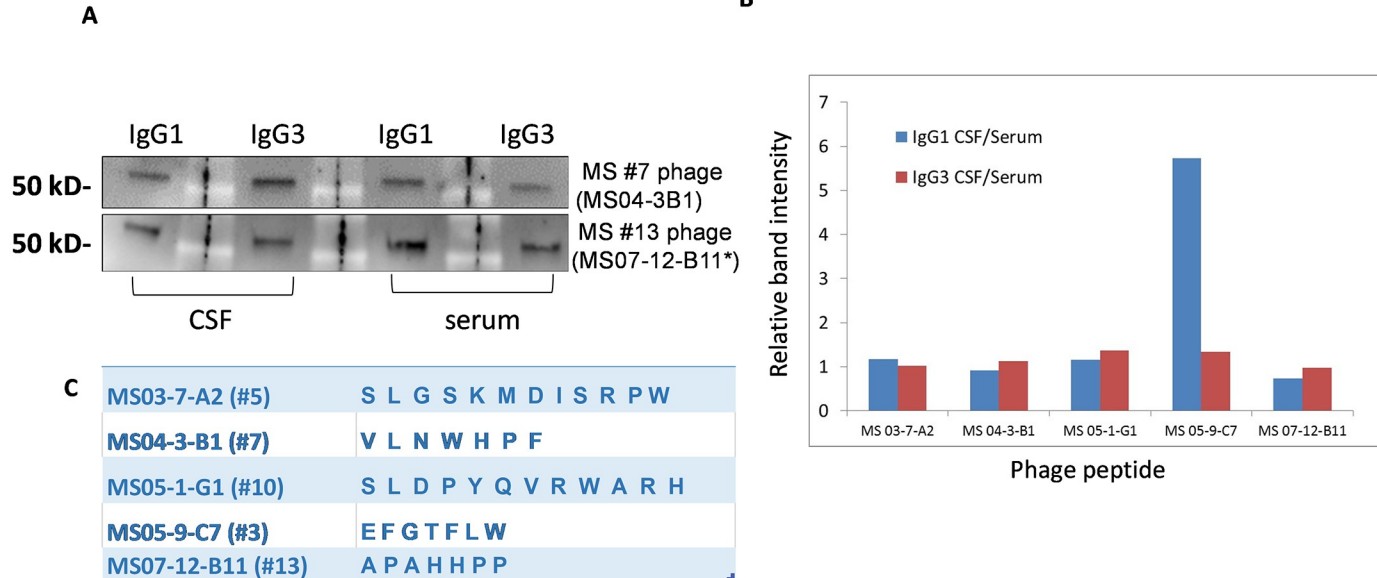

**Fig 3. Western blots show that the intrathecal IgG-reactive phage peptides were recognized by both IgG1 and IgG3 subclasses MS CSF and serum. A.**
SDS-PAGE Western blots of phage peptides show that they were reactive to both IgG1 and IgG3 subclasses in paired CSF and serum of MS patients. Phage peptides
($10^{10}$/well) were separated on a 4–15% SDS-PAGE gel, blotted, and probed with corresponding total CSF and serum (primary antibody) of MS patients from which
the original phage peptides were identified. The bound CSF and serum antibodies were then probed with mouse anti-human IgG1 and IgG3 antibodies, followed by
anti-mouse-HRP antibodies and detected with SuperSignal® West Femto chemiluminescent substrate. Representative phage shown here are phage peptide from
MS #7 (MS 04–3 B1) and phage peptide from MS #13 (MS07-12 B11). **B**. Band intensity analysis of phage Western blots with IgG1/IgG3 probes showed that the
ratio of IgG1 band between CSF and serum was similar as ratio of IgG3 band between CSF and serum. C. Peptides used for the western blots. *same as peptide
selected by #13 (MS05-4) [10].

sclerosing panencephalitis, acute disseminated encephalomyelitis, neurosyphilis, sarcoid, VZV
myelopathy and radiculomyelitis) with representative phage peptides. Patient CSF characteristic including IgG concentration, diagnosis, and OCBs are included in Supplemental data (S1
and S2 Tables). With highly sensitive and specific phage mediated real-time IPCR, we failed to
identify any binding reactivity of the peptides as common antigens to MS CSF nor IC CSF (S1
Fig), suggesting that these peptides are patient-specific.

## Protein database searches with selected MS peptides

Representative peptides were used to search for homologous sequences in the protein database
(https://blast.ncbi.nlm.nih.gov/Blast) to identify corresponding protein candidates. Because of
the short length of the peptide sequences, numerous candidate proteins were revealed including proteins of viral, bacterial and other origins. Once again, no common shared protein/peptides were identified. Table 3 shows a list of candidate proteins from human and bacteria. The
most abundant candidates are proteins involved in cellular stress, transcription factors, membrane proteins, neuronal proteins, enzymes, and immunoglobulin fragments with peptide
sequence identities ranged 62%–100%).

## Discussion

We previously showed that there is a temporal stability of peptide antigens for the intrathecal
IgG antibodies in the CSF of patients with MS, suggesting the importance of applying a
phage peptide approach to identify targets of the intrathecal IgG [10]. In this study, we have
expanded the sample size for screening which includes CSF from 20 OCB-positive MS patients

**Table 3. Protein database search results with selected MS peptides.**

| Peptide/Patient ID | Homology to known proteins (Species) | Identities | Sequence ID |
|---|---|---|---|
| S T L S E S K V N R L L | >molecular chaperone DnaK (Euhalothece sp.) | 9/12(75%) | PNW58714.1 |
| (MS02-19) | >zinc finger, DHHC-type containing 20, isoform CRA_d (human) | 9/10(90%) | EAX08310.1 |
| K F G T A L W | >ABC transporter permease (Butyrivibrio sp.) | 6/7(86%) | WP_026507654.1 |
| (MS02-24) | >neurofascin isoform X1 (human) | 6/6(100%) | XP_024310051.1 |
| S L G S K M D I S R P W | >GTP-binding protein (unclassified Staphylococcus) | 9/12(75%) | WP_070485309.1 |
| (MS03-7) | > HSPC019 (human) | 7/8(88%) | DAA00039.1 |
| N P V E H W L A V L P T | >ATPase, partial (Micromonospora sp.) | 8/9(89%) | WP_109816073.1 |
| (MS04-2) | >immunoglobulin heavy chain variable region, partial (human) | 6/6(100%) | AIZ70805.1 |
| V L N W H P F | >molecular chaperone DnaJ (Escherichia fergusonii) | 6/6(100%) | WP_046083568.1 |
| (MS04-3) | >NADPH oxidase 3 (human) | 5/6(83%) | NP_056533.1 |
| S L D P Y Q V R W A R H | >alpha-amylase (Pseudopropionibacterium propionicum) | 8/9(89%) | WP_061787838.1 |
| (MS05-1) | >N-acetylglucosaminyltransferase (human) | 8/13(62%) | SJM34704.1 |
| A T L T A A T S G S T V | >carbohydrate ABC transporter permease (Bacillus horikoshii) | 9/11(82%) | WP_088019463.1 |
| (MS05-2) | >immunoglobulin heavy chain variable region, partial (human) | 9/11(82%) | BAI51901.1 |
| I P Y H R F P | >acetylornithine deacetylase (unclassified Bosea) | 6/7(86%) | WP_114828852.1 |
| (MS05-3) | >mucin 2, oligomeric mucus/gel-forming, isoform CRA_a (human) | 5/5(100%) | EAX02421.1 |
| A P A H H P P | >glycosyltransferase (Streptomyces rimosus) | 7/7(100%) | WP_003986546.1 |
| (MS05-4) | >zinc-finger homeodomain protein 4 (human) | 6/6(100%) | BAE96598.1 |
| L I S I S E Q R A A L I | >nuclear pore complex protein nup155 (Hymenolepis microstoma) | 9/11(82%) | CDS29450.1 |
| (MS05-7) | >regulating synaptic membrane exocytosis protein 2 isoform X1 (human) | 7/9(78%) | XP_011515697.1 |
| G W T H F D K P I G T L | >ABC transporter substrate-binding protein (Ochrobactrum sp.) | 8/11(73%) | WP_095444449.1 |
| (MS05-8) | >immunoglobulin E heavy chain variable region, partial (human) | 9/13(69%) | ACZ04682.1 |
| F Y S H S F P P | >autotransporter domain-containing protein (Nonlabens arenilitoris) | 7/8(88%) | WP_105070814.1 |
| (MS06-3) | >Atrophin 1 (human) | 6/7(86%) | AAH51795.1 |
| Y Y P F T S M G P A Q S | >T-cell leukemia homeobox protein 3-like (Limulus Polyphemus) | 10/14(71%) | XP_013794488.1 |
| (MS07-11) | >interferon, gamma-inducible protein 16 (human) | 8/11(73%) | EAW52802.1 |

Exemplary peptides were used to search for homologous sequences in the protein database (https://blast.ncbi.nlm.nih.gov/Blast.cgi?PAGE=Proteins). Listed are 2 examples from aligned proteins (one from bacteria, and one from human).

for panning phage-displayed random peptide libraries. A total of 40 unique, specific peptides were identified by 14 MS CSF, and no peptide sequences were shared among the peptides identified. We demonstrated that these phage peptides were targeted by intrathecal oligoclonal IgG antibodies/oligoclonal bands. Furthermore, these peptides reacted to both IgG1 and IgG3 subclasses. Phage mediated immuno-PCR screening of 42 MS and 13 inflammatory control CSF revealed that no MS specific peptide antigens were found. We postulate that the oligoclonal IgG antibodies in MS may target patient specific antigens, and that the significance of OCB may not due to the antigens specific to antibodies in MS.

Oligoclonal bands can be detected in the CSF of most (>95%) patients with MS, persist throughout the course of the disease, and are a diagnostic hallmark of the disease [14]. The critical role of intrathecal IgG/oligoclonal bands in MS disease pathogenesis is supported by mounting evidence. For example, actively demyelinating lesions are commonly associated with prominent deposition of immunoglobulins and complement activation products [15–18], OCBs are shown to be associated with increased levels of disease activity and disability [1,3,6], a greater risk of second attack [19], the conversion from a clinically isolated syndrome (CIS) to early RRMS [4,20], and greater brain atrophy [5,6]. Moreover, the presence of OCB in CSF in the specific clinical context is still the most reliable parameter to confirm the likely diagnosis of MS [21,22], supporting the critical pathological role of intrathecal IgG antibodies in MS.

Intrathecal production of antibodies against viruses (measles, rubella, and varicella zoster virus), bacteria and CNS components in MS have up until now shown inconsistent or negative results [23–25]. Further, recombinant antibodies generated from clonally expanded B cells/plasma cells in the CSF and from laser capture microdissection of B cells in MS lesions failed to identify MS specific antigens [26,27]. A recent report showed that OCB in MS target ubiquitous intracellular antigens released in cellular debris [9].

We used a combination of two phage peptide libraries (Ph.D. 12 and Ph.D. 7) and extensive panning strategies [11–13] with increased rounds of panning (up to six rounds) and stringency, and identified 40 specific phage peptides from 70 percent of the MS CSF (14 out of 20) used. Significantly, these phage peptides were recognized by intrathecal IgG/oligoclonal bands as demonstrated by our techniques of phage-probed IEF and phage-mediated immune-PCR, suggesting that phage peptides can be a unique tool for investigating antigen specificity of MS oligoclonal bands which may give critical clues as to the disease causation. Despite the high specificity of the intrathecal IgG specific phage peptides, the highly sensitive and specific phage-mediated immuno-PCR technique used here failed to identify common peptide reactivity shared by all MS CSF screened, implying that MS intrathecal IgG antibodies may target patient specific antigens. Our data are consistent with previous studies and support the notion that the disease targets for OCBs in MS have yet to be reproducibly demonstrated [28,29]. Failing to identify MS specific targets for OCBs does not diminish its crucial roles in disease pathogenesis, as the presence of large amount of IgG antibodies are key features of MS lesions [30,31], and have been shown to be associated with increased disease activity and brain atrophy [3,6]. Over 20 times more IgG were extracted from MS plaques than those from control brain [32]. Furthermore, significant higher amounts of bound IgG (oligoclonal in nature) were eluted from MS brains with both high and low pH buffer [30,32,33], and the consistent presence of complement, IgG antibodies, and Fcγ receptors (FcR) in phagocytic macrophages suggests that antibody- and complement-mediated myelin phagocytosis is the dominant mechanism of demyelination in established MS [34]. We have previously analyzed clinical laboratory parameters from 91 patients with MS and showed that in MS there was not a linear relationship between the numbers of OCBs and CSF IgG concentration [35]. The complex relationship between OCBs and other CSF parameters suggests that at certain concentrations, the IgG antibody in the CNS is being sequestered or aggregated to form IgG complexes (as bound IgG) and therefore unable to contribute to the number of OCBs, with the relationships become negatively associated [35].

IgG1 and IgG3 are the first 2 Ig classes after IgM. They have superior ability to activate effector functions. Both IgG1 and IgG3 subclasses were found to be present in the same OCB in MS CSF [36,37], and the elevation of IgG1 and IgG3 indices in MS were found more frequently than the elevation of the general IgG index [1]. Furthermore, patients with a relapse were significantly more frequently seropositive for anti-MOG and anti-MBP IgG3 than those in remission [38]. It would be interesting to investigate IgA, IgG2 and IgG4 in MS but we did not carry out these experiments due to the limitation of study scope. Using western blots analysis, we showed that the intrathecal phage peptides were recognized by IgG1 and IgG3 antibodies in both CSF and paired serum of MS patients, with equal band intensities, indicating that both subclasses could be important for disease. Both IgG1 and IgG3 subclasses have been found in the intrathecal IgG [39] and in the same oligoclonal bands [36], indicating that the OCBs may be consisted of at least IgG1 and IgG3 subclasses. Further, molecular sequencing data revealed that in MS clonally related and even the same IgG-VH sequences are found in multiple OCBs band [40], further supporting the notion that OCBs may be represented by multiple IgG subclasses.

The distinct sets of oligoclonal IgG-reactive peptides identified by individual MS CSF suggest that the elevated intrathecal antibodies may target patient-specific antigens, it also indicates the limitation of this approach. Other limitations include the relatively short nature of the linear phage display peptides that limits recognition of more extensive secondary or tertiary structures/epitopes. The discovered "mimotopes" might not be the real antigens, which prevents us from discovering concordant antibody binding between patients. Other advance technologies might be more successful. Nonetheless, further investigations are needed to determine characteristics and the role of the increased intrathecal antibodies in MS.

## Supporting information

**S1 Fig. A**. **There is no shared and differential phage peptide binding between MS and IC CSF**. IPCR was performed to screen 42 MS and 13 IC CSF with pooled phage peptides MS05-4A6, MS05-4A2, and MS04-3B1. p = 0.47. **B**. **Summary of S1 Fig**. **A**. **There is no shared and differential phage peptide binding between MS and IC CSF**.
(TIF)

**S1 Table. MS CSF used for IPCR screening.**
(DOCX)

**S2 Table. Inflammatory control (IC) patients used for IPCR CSF screening.**
(DOCX)

**S1 Raw Images. Original images for IEF and western blots.**
(PDF)

## Acknowledgments

We dedicate this paper to Dr. Don Gilden for his lifelong passion and effort in trying to find the cause of MS.

## Author Contributions

**Conceptualization:** Timothy Vollmer, Xiaoli Yu.

**Data curation:** Tiffany Pointon, Sean Manton, Miyoko Green, Kathryn Dennison, Mollie Davis, Gino Braiotta, Taylor Edwards, Bailey Polonsky.

**Funding acquisition:** Xiaoli Yu.

**Investigation:** Michael Graner.

**Methodology:** Tiffany Pointon.

**Project administration:** Xiaoli Yu.

**Resources:** Timothy Vollmer.

**Supervision:** Xiaoli Yu.

**Visualization:** Anthony Fringuello.

**Writing – original draft:** Michael Graner.

**Writing – review & editing:** Julia Craft, Anthony Fringuello, Xiaoli Yu.

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
