## [Decision Letter · Decision Letter 0]

15 Nov 2019

PONE-D-19-24317

Oligoclonal IgG Antibodies in Multiple Sclerosis Target Patient-Specific Peptides

PLOS ONE

Dear Dr Xiaoli Yu

Thank you for submitting your manuscript to PLOS ONE. After careful consideration, we feel that it has merit but does not fully meet PLOS ONE’s publication criteria as it currently stands. Therefore, we invite you to submit a revised version of the manuscript that addresses the all points raised during the review process.

We would appreciate receiving your revised manuscript by 3 months. To enhance the reproducibility of your results, we recommend that if applicable you deposit your laboratory protocols in protocols.io, where a protocol can be assigned its own identifier (DOI) such that it can be cited independently in the future. For instructions see: http://journals.plos.org/plosone/s/submission-guidelines#loc-laboratory-protocols

We look forward to receiving your revised manuscript.

Kind regards,

Brahim Nait-Oumesmar

Academic Editor

PLOS ONE

Journal Requirements:

1. Please provide additional details regarding participant consent. In the ethics statement in the Methods and online submission information, please ensure that you have specified (1) whether consent was informed and (2) what type you obtained (for instance, written or verbal, and if verbal, how it was documented and witnessed). If your study included minors, state whether you obtained consent from parents or guardians. If the need for consent was waived by the ethics committee, please include this information

2.

PLOS ONE now requires that authors provide the original uncropped and unadjusted images underlying all blot or gel results reported in a submission’s figures or Supporting Information files. This policy and the journal’s other requirements for blot/gel reporting and figure preparation are described in detail at https://journals.plos.org/plosone/s/figures#loc-blot-and-gel-reporting-requirements and https://journals.plos.org/plosone/s/figures#loc-preparing-figures-from-image-files. When you submit your revised manuscript, please ensure that your figures adhere fully to these guidelines and provide the original underlying images for all blot or gel data reported in your submission. See the following link for instructions on providing the original image data: https://journals.plos.org/plosone/s/figures#loc-original-images-for-blots-and-gels.

**Comments to the Author**

1. Is the manuscript technically sound, and do the data support the conclusions?

Reviewer #1: Yes

Reviewer #2: Yes

2. Has the statistical analysis been performed appropriately and rigorously? 

Reviewer #1: No

Reviewer #2: Yes

3. Have the authors made all data underlying the findings in their manuscript fully available?

Reviewer #1: Yes

Reviewer #2: Yes

4. Is the manuscript presented in an intelligible fashion and written in standard English?

Reviewer #1: Yes

Reviewer #2: Yes

5. Review Comments to the Author

Reviewer #1: This is an interesting paper. The following should be addressed:

Lines 21, 144, 266, 294, and 299: The authors can’t claim “high affinity” –affinity was not tested.

The authors can’t emphasize that “no shared peptide sequences were found among MS patients” (lines 160, 267), as MS#10 and #11 share peptide ATLTAATSGSTV (Table 2).

Line 171: *peptides were published previously (Yu 2011): Were the same CSF samples screened again for antigenic peptides? Or re-analyzed? Clarification is needed. Peptides do not overlap 1:1 with the 2011 publication, as additional peptides are added. Some patients, e.g. MS04-3, MS05-6 have asterisks in table 2, but were not listed in 2011.

Fig. 1 needs statistics. Is the difference between CSF and serum significant? Which test was used?

Lane 181: “Phage peptides selected by CSF IgG of all 14 MS patients bound more to CSF IgG than to serum IgG of the same patient” -> but Fig. 1 shows only results from 2 patients -> if tested, additional data must be listed.

Fig. 1: Both patients, MS02-19 and MS03-7 were already investigated in Yu 2011. The same experiment for the same patients has been shown in Yu, 2011, Fig. 7. Clarify what is new!

Fig. 2: the Anti-HIgG-stained CSF samples don’t look oligoclonal but rather polyclonal, however it could be due to low resolution of the pre-prints. Their claim that “peptide-specific OCBs correspond to some of the major bands in the OCB pattern of the MS patients detected by anti-human IgG” (Figure legend 2, lane 213), cannot be corroborated by the low-resolution images.

Fig. 3: Why was IgG1 and IgG3 chosen? Were other classes tried as well? If so, please mention. IgG3 and IgG1 are the first 2 Ig classes after IgM. Even more interesting would be IgA, IgG2 and IgG4, to check if class-switch occurs towards IgA and beyond and to check if mucosal-associated B cells (IgA) traffick to the CSF space. -> Discussion, line 325 onwards, discuss class switching and relevance/properties of Ig-classes.

Fig. 3: Line 220: the authors claim “representative blots” from patients MS #7 and #13, but Fig. 3A shows only representative plots from MS #07.

Did they test all phage peptides with CSF and Serum of all patients? Or only the 5 peptides that are listed in Fig. 3B? Which peptides are those? Abbreviations (A3, B1, G1…) do not correspond to any reference within the manuscript.

Are the anti-IgG1 and anti-IgG3 antibodies isotype-specific and cross-absorbed? The company and catalogue-number should be listed in the methods section.

Table 3: species should be listed for each protein.

Discussion: Multiple grammatical errors and incomplete sentences. Revise.

Line 32-34 and 272-274: the paper does not deliver any evidence for the existence and pathogenicity of immune complexes. The speculation should be taken out of the abstract and a reference must be provided in the discussion. Lines 336 onwards: The authors mention IgG1-IgG3 immune complexes, but are the complexes specific to these classes? Do they want to emphasize lack of low-immunogenic IgG4? No evidence is presented for it. Less emphasis should be spent on the immune complex hypothesis, but rather on the specific peptides they present and on the relevance of their corresponding proteins.

Lines 305-309: Instead of this bold claim, the authors should discuss the limitations of their phage display approach: the discovered “mimotopes” might not be the real antigens, which prevents them from discovering concordant antibody binding between patients. Phage display is a relevant method, but only one of many and others might be more successful.

Minor corrections:

Line 26: “viral” origin – no viral proteins are discussed in detail in the manuscript.

Methods paragraph line 70-77 should contain basic information on Ph.D phage displays: How many sequences included? Random sequences? Peptide length? Were both displays used for each sample?

Line 167: RPMS: should rather be called SPMS. Patient 3 is already called SPMS in the table.

Lane 201: “oligoclonal clonal”

Lane 241: If possible list MS and control patients in table with additional characteristics (µg/ml IgG in CSF, OCB present in those MS patients, too?)

Line 244: Were all described phage peptides tested? If possible, show data in suppl. figure together with positive controls.

Line 257: replace “representative” with “exemplary”.

Line 264: incomplete sentence.

Line 287: incomplete sentence.

Line 310: singular – plural

Line 314: significantly

Line 325: western blot analysis.

Line 327, 332, revise.

Reviewer #2: Strengths: The investigators aim was to examine the antigen specificity of 20 MS patients whose CSF showed the presence of oligoclonal IgG bands (OCB), and identify 40 high affinity unique peptides by panning phage-displayed random peptide libraries. Here the authors have extended their preliminary studies to investigate the specificity of OCBs. The data showed that within individual MS patients, there was no shared peptide sequences were found in 42 MS or 13 controls. The aim is clearly focused, and experimental work is well carried out. The results are well presented. Although the rationale of the work has been clearly described, the authors did not write a clear hypothesis.

Weaknesses: The reviewer is wondering whether the authors have examined the peptide antigens in individual MS patients during disease activity. Longitudinal studies in individual MS patients might provide us better understanding of the specificity of OCBs contributing to disease pathogenesis. Page 16: The authors postulated that the significance of OCBs may be due to the elevated amount of antibodies which form immune complexes. Here the authors have presented no preliminary data. The discussion is too long, and not relevant to the present study. My recommendation is to publish the article in a form of short communication.

6. PLOS authors have the option to publish the peer review history of their article (what does this mean?). If published, this will include your full peer review and any attached files.

Reviewer #1: No

Reviewer #2: Yes: Mehta Pankaj D

---

## [Author Response · Author response to Decision Letter 0]

10 Dec 2019

Response to Reviewers 

We thank the reviewers for the critical comments. Our responses are listed after each questions/comments.

Reviewer #1:

Lines 21, 144, 266, 294, and 299: The authors can’t claim “high affinity” –affinity was not tested.

In the revised manuscript, we replaced “high affinity” with words of “specificity” or “specific” (lines 21, 151, 284, 313, and 318).

The authors can’t emphasize that “no shared peptide sequences were found among MS patients” (lines 160, 267), as MS#10 and #11 share peptide ATLTAATSGSTV (Table 2).

It was a mistake to include the peptide ATLTAATSGSTV selected by MS#11 into the peptide list of MS#10. We removed this peptide from the list of MS#10 in the revised manuscript. 

Line 171: *peptides were published previously (Yu 2011): Were the same CSF samples screened again for antigenic peptides? Or re-analyzed? Clarification is needed. Peptides do not overlap 1:1 with the 2011 publication, as additional peptides are added. Some patients, e.g. MS04-3, MS05-6 have asterisks in table 2, but were not listed in 2011.

The same CSF samples published previously (Yu 2011) were not screened again but were re-analyzed. Peptides selected by MS04-3 and MS05-6 were published previously in our publication “Intrathecally synthesized IgG in multiple sclerosis cerebrospinal fluid recognizes identical epitopes over time” (Yu et al., 2011), but an incorrect paper was cited as there were several papers by Dr. Yu in 2011. The correct reference (Yu 2011) was included in the revised manuscript (line 413). 

Fig. 1 needs statistics. Is the difference between CSF and serum significant? Which test was used?

Yes, there is a significant difference between CSF and serum (p=0.0002) (line 185-189). Paired Student’s T-Test was used (line 198-201).

Lane 181: “Phage peptides selected by CSF IgG of all 14 MS patients bound more to CSF IgG than to serum IgG of the same patient” -> but Fig. 1 shows only results from 2 patients -> if tested, additional data must be listed.

We included a new phage IPCR figure (Fig. 1C) containing all phage peptides tested and demonstrated that there is significantly higher phage peptide binding to MS CSF IgG compared to paired serum IgG. 

Fig. 1: Both patients, MS02-19 and MS03-7 were already investigated in Yu 2011. The same experiment for the same patients has been shown in Yu, 2011, Fig. 7. Clarify what is new!

In the current manuscript, we showed that phage peptides selected by MS02-19 and MS03-7 were recognized by intrathecal IgG by phage-mediated immune PCR (Fig.1) and by IEF immunoblotting which were not shown in our previous publication. 

Fig. 2: …the Anti-HIgG-stained CSF samples don’t look oligoclonal but rather polyclonal, however it could be due to low resolution of the pre-prints. Their claim that “peptide-specific OCBs correspond to some of the major bands in the OCB pattern of the MS patients detected by anti-human IgG” (Figure legend 2, lane 213), cannot be corroborated by the low-resolution images.

We tried and failed to improve the resolution of the IEF blots. In the revised manuscript, the sentence “peptide-specific OCBs correspond to some of the major bands in the OCB pattern of the MS patients detected by anti-human IgG” was deleted (line 216-217). 

Fig. 3: Why was IgG1 and IgG3 chosen? Were other classes tried as well? If so, please mention. IgG3 and IgG1 are the first 2 Ig classes after IgM. Even more interesting would be IgA, IgG2 and IgG4, to check if class-switch occurs towards IgA and beyond and to check if mucosal-associated B cells (IgA) traffick to the CSF space. -> Discussion, line 325 onwards, discuss class switching and relevance/properties of Ig-classes.

We did not test other IgG subclasses. We chose IgG 1 and IgG 3 due to the following evidence. IgG3 and IgG1 are the first 2 next Ig subclasses after IgM, with different Fc receptor binding and affinities. They have superior ability to activate effector functions such as C1q binding for complement activation. Both IgG1 and IgG3 subclasses were found to be present in the same OCB in MS CSF (Grimaldi et al., 1986, Losy J, Mehta, et al., 1990), and the elevation of IgG1 and IgG3 indices in MS were found more frequently than the elevation of the general IgG index (Caroscio et al., 1986). Furthermore, patients with a relapse were significantly more frequently seropositive for anti-MOG and anti-MBP IgG3 than those in remission (Garcia-Merino et al., 1986). It would be interesting to investigate IgA, IgG2 and IgG4 in MS but we did not carry out these experiments due to the limitation of the study scope (line 344-350).

Fig. 3: Line 220: the authors claim “representative blots” from patients MS #7 and #13, but Fig. 3A shows only representative plots from MS #07.

Did they test all phage peptides with CSF and Serum of all patients? Or only the 5 peptides that are listed in Fig. 3B? Which peptides are those? 

Abbreviations (A3, B1, G1…) do not correspond to any reference within the manuscript.

A representative blot from patient MS #13 is included in the revised manuscript. Only the phage peptides listed in Fig. 3B were tested. Peptide sequences used for the western blots were included (Fig. 3C) in the revised manuscript. 

Are the anti-IgG1 and anti-IgG3 antibodies isotype-specific and cross-absorbed? The company and catalogue-number should be listed in the methods section.

Isotype-specific mouse monoclonal anti-human IgG1 (I2513, clone 8c/6-39) and anti-human IgG3 (I7260, clone HP-6050) antibodies were used (Sigma) (line 128-130).

Table 3: species should be listed for each protein.

Species are included in Table 3 in the revised manuscript. 

Discussion: Multiple grammatical errors and incomplete sentences. Revise.

We revised and shortened the discussion (per Reviewer 2’s request as well). 

Line 32-34 and 272-274: the paper does not deliver any evidence for the existence and pathogenicity of immune complexes. The speculation should be taken out of the abstract and a reference must be provided in the discussion. Lines 336 onwards: The authors mention IgG1-IgG3 immune complexes, but are the complexes specific to these classes? Do they want to emphasize lack of low-immunogenic IgG4? No evidence is presented for it. Less emphasis should be spent on the immune complex hypothesis, but rather on the specific peptides they present and on the relevance of their corresponding proteins.

The speculation of “Immune complexes” and related sentences were removed from the revised manuscript (line 31-34; 353-354; 356; 359-360; 369-379). In the revised manuscript, we eliminated or lessoned downplayed the immune complex hypothesis, and emphasized on the specific peptides and the relevance of their corresponding proteins.

Lines 305-309: Instead of this bold claim, the authors should discuss the limitations of their phage display approach: the discovered “mimotopes” might not be the real antigens, which prevents them from discovering concordant antibody binding between patients. Phage display is a relevant method, but only one of many and others might be more successful.

The points are well taken. The revised manuscript includes a discussion on the limitation of our approach (line 347-352). 

Minor corrections:

Line 26: “viral” origin – no viral proteins are discussed in detail in the manuscript.

In the revised manuscript, we added “including proteins of viral, bacterial and other species (line 262). 

Methods paragraph line 70-77 should contain basic information on Ph.D phage displays: How many sequences included? Random sequences? Peptide length? Were both displays used for each sample?

The revised manuscript includes basic information of Phage-display libraries used (line 72-76).

Line 167: RPMS: should rather be called SPMS. Patient 3 is already called SPMS in the table.

In the revised manuscript, SPMS replaced RPMS. 

Lane 201: “oligoclonal clonal”

In the revised manuscript, the mistake was corrected (line 209). 

Lane 241: If possible list MS and control patients in table with additional characteristics (µg/ml IgG in CSF, OCB present in those MS patients, too?)

MS and control patients were included in Supplemental data. Table 1 lists MS patients used for phage peptide screening and table 2 contains inflammatory control patients used for phage peptide screening. 

Only representative phage peptides MS02-24-A6, MS04-3-B1, and MS05-4-A2 were tested. Data are shown in Supplemental data Fig. 1A and 1B. 

Line 257: replace “representative” with “exemplary”.

Line 264: incomplete sentence.

Line 287: incomplete sentence.

Line 310: singular – plural

Line 314: significantly

Line 325: western blot analysis.

Line 327, 332, revise.

All these pointes were addressed in the revised manuscript. 

Reviewer #2:

Although the rationale of the work has been clearly described, the authors did not write a clear hypothesis.

In the revised manuscript, we added “We hypothesize that phage-displayed random peptide libraries can be used to identify antigenic peptides specific to intrathecal IgG of MS” in the introduction (line 51-52). 

The reviewer is wondering whether the authors have examined the peptide antigens in individual MS patients during disease activity. Longitudinal studies in individual MS patients might provide us better understanding of the specificity of OCBs contributing to disease pathogenesis. 

We agree. Longitudinal studies in individual MS patients of antigenic peptides to OCBs were examined and published (Intrathecally synthesized IgG in multiple sclerosis cerebrospinal fluid recognizes identical epitopes over time) (Yu et al., 2011). We showed that peptides recognized multiple IgG bands in the longitudinal MS CSFs and importantly, more OCBs were revealed in the second CSF, suggesting that these peptides are valuable tools for us to understand OCB contribution to disease pathogenesis (Fig. 2, Yu et al., 2011). 

Page 16: The authors postulated that the significance of OCBs may be due to the elevated amount of antibodies which form immune complexes. Here the authors have presented no preliminary data. The discussion is too long, and not relevant to the present study. My recommendation is to publish the article in a form of short communication.

In the revised manuscript, we eliminated “immune complexes” and all related sentences. We shortened the discussion and remove sentences regarding our postulations. As PLOS ONE does not have word count or page length restrictions, it is not clear that defined short communications are in the repertoire of types of articles the journal publishes. Thus, we feel that a full-length manuscript would present our data more clearly instead of a short communication.

---

## [Decision Letter · Decision Letter 1]

27 Jan 2020

Oligoclonal IgG Antibodies in Multiple Sclerosis Target Patient-Specific Peptides

PONE-D-19-24317R1

Dear Dr.Xiaoli,

We are pleased to inform you that your manuscript has been now judged scientifically suitable for publication and will be formally accepted for publication once it complies with all outstanding technical requirements.

With kind regards,

Brahim Nait-Oumesmar

Academic Editor

PLOS ONE

Additional Editor Comments (optional):

Reviewers' comments:

Reviewer's Responses to Questions

**Comments to the Author**

1. If the authors have adequately addressed your comments raised in a previous round of review and you feel that this manuscript is now acceptable for publication, you may indicate that here to bypass the “Comments to the Author” section, enter your conflict of interest statement in the “Confidential to Editor” section, and submit your "Accept" recommendation.

Reviewer #1: All comments have been addressed

Reviewer #2: All comments have been addressed

2. Is the manuscript technically sound, and do the data support the conclusions?

Reviewer #1: Yes

Reviewer #2: (No Response)

3. Has the statistical analysis been performed appropriately and rigorously? 

Reviewer #1: Yes

Reviewer #2: (No Response)

4. Have the authors made all data underlying the findings in their manuscript fully available?

Reviewer #1: Yes

Reviewer #2: (No Response)

5. Is the manuscript presented in an intelligible fashion and written in standard English?

Reviewer #1: Yes

Reviewer #2: (No Response)

6. Review Comments to the Author

Reviewer #1: All comments have been addressed in a satisfactory manner. The comments all were reviewed and answered with care. The manuscript is a valuable addition to our understanding of the specficity and diversity of the immune response in MS.

Reviewer #2: All my review comments have been adequately addressed by the authors, therefore the manuscript is now acceptable for publication.

7. PLOS authors have the option to publish the peer review history of their article (what does this mean?). If published, this will include your full peer review and any attached files.

Reviewer #1: Yes: Lawrence Steinman

Reviewer #2: Yes: Mehta Pankaj D

---

## [Editor Report · Acceptance letter]

13 Feb 2020

PONE-D-19-24317R1 

Oligoclonal IgG Antibodies in Multiple Sclerosis Target Patient-Specific Peptides 

Dear Dr. Yu:

I am pleased to inform you that your manuscript has been deemed suitable for publication in PLOS ONE. Congratulations! Your manuscript is now with our production department. 

With kind regards,

on behalf of

Dr. Brahim Nait-Oumesmar 

Academic Editor

PLOS ONE